# Exosomes: Diagnostic and Therapeutic Implications in Cancer

**DOI:** 10.3390/pharmaceutics15051465

**Published:** 2023-05-11

**Authors:** Hyein Jo, Kyeonghee Shim, Dooil Jeoung

**Affiliations:** Department of Biochemistry, College of Natural Sciences, Kangwon National University, Chuncheon 24341, Republic of Korea

**Keywords:** cellular interactions, clinical trials, diagnostics, drug carriers, exosomes, miRNAs, prognostics, siRNAs, therapeutics

## Abstract

Exosomes are a subset of extracellular vesicles produced by all cells, and they are present in various body fluids. Exosomes play crucial roles in tumor initiation/progression, immune suppression, immune surveillance, metabolic reprogramming, angiogenesis, and the polarization of macrophages. In this work, we summarize the mechanisms of exosome biogenesis and secretion. Since exosomes may be increased in the cancer cells and body fluids of cancer patients, exosomes and exosomal contents can be used as cancer diagnostic and prognostic markers. Exosomes contain proteins, lipids, and nucleic acids. These exosomal contents can be transferred into recipient cells. Therefore, this work details the roles of exosomes and exosomal contents in intercellular communications. Since exosomes mediate cellular interactions, exosomes can be targeted for developing anticancer therapy. This review summarizes current studies on the effects of exosomal inhibitors on cancer initiation and progression. Since exosomal contents can be transferred, exosomes can be modified to deliver molecular cargo such as anticancer drugs, small interfering RNAs (siRNAs), and micro RNAs (miRNAs). Thus, we also summarize recent advances in developing exosomes as drug delivery platforms. Exosomes display low toxicity, biodegradability, and efficient tissue targeting, which make them reliable delivery vehicles. We discuss the applications and challenges of exosomes as delivery vehicles in tumors, along with the clinical values of exosomes. In this review, we aim to highlight the biogenesis, functions, and diagnostic and therapeutic implications of exosomes in cancer.

## 1. Introduction

For this review, we first wanted to search papers concerning exosomes. More than 27,000 publication records were obtained through a PubMed search. A literature survey to identify papers describing the properties and functions of exosomes was first conducted in PubMed on 5 December 2022. For this review, we wanted to include both research papers and review papers. More than 90% of the papers in this review were published in the last 5 years. After the removal of non-English publications, 163 publications were screened based on their title and abstract. For some papers (31/163), PDFs were not available. All of the papers in this review are intended to enhance the understanding clinical values of exosomes, including biogenesis, functions, diagnostic and/or prognostic markers, and provide examples in drug delivery and therapy.

Extracellular vesicles (EVs) comprise three groups: (i) exosomes, (ii) microvesicles (MVs), and (iii) apoptotic bodies [1]. Molecules in these EVs mediate signaling pathways in tumor, immune, and stromal cells to promote cancer initiation and progression. The fusion of multivesicular bodies (MVBs) with the plasma membrane leads to the secretion of exosomes, which have a diameter of 30~150 nm [2,3,4]. Exosomes are surrounded by a lipid bilayer membrane [5], and they were originally thought to be cellular waste products [6].

Exosomes are present in culture supernatants, serum, plasma, urine, and various other sources [7]. They contain DNA, miRNAs, long non-coding RNAs (lnc RNAs), circulatory RNAs (circ RNAs), DNA fragments, proteins, and various lipids [8,9,10].

Exosomes mediate cell-to-cell communication [9,11], and exosomal contents mediate the effects of exosomes on various life processes. Exosomes are involved in angiogenesis [12], metastasis [13,14,15], anticancer drug resistance [16], immune evasion [17], macrophage polarization [18], and metabolic reprogramming [19]. Altogether, these reports suggest that exosomes may contribute to the pathogenesis of diseases.

## 2. Biogenesis of Exosomes

Since exosomes play important roles in various life processes, it is likely that exosomes can serve as targets for developing anticancer drugs. It is therefore important to understand the mechanisms associated with the biogenesis of exosomes. Exosomes originate from endocytosis [20]. Endocytosis occurs via dynamin-dependent or dynamin-independent pathways. Dynamin-dependent pathways include the clathrin-dependent and endophiln-A2-dependent pathways [20], whereas dynamin-independent pathways include the c-terminal binding protein 1 (CTBP1)-dependent pathway [21] (Figure 1), phagocytosis, and micropinocytosis [21,22,23]. Caveolin-1, an integral membrane protein, initiates caveola on the plasma membrane and promotes caveola-dependent endocytosis [24] (Figure 1). Endophilin-A2 mediates the endocytosis of membrane proteins [25] (Figure 1).

The formation of exosomes comprises the following steps [26,27] (Figure 1): (1) the invagination of the plasma membrane to form early endosomes, (2) the formation of intraluminal vesicles (ILVs) contained within MVBs, (3) the maturation of late endosomes (MVBs) via acidification, and (4) the extracellular release of ILVs as exosomes through fusion with the plasma membrane. The initial endocytic vesicles are fused via caveolin-dependent or caveolin-independent mechanisms, and this serves as the first step in developing early endosomes [28,29]. Early endosomal markers include the ras family of GTPase 4 (RAB4), RAB5A, transferrin and its receptor, and early endosome antigen 1 [28,29] (Figure 1). The formation of ILVs within the endosome is enhanced by lipid Bis (monoacylglycero)phosphate or lysobisphosphatidic acid, which are only found in late endosomes, endolysosomes, or lysosomes. The formation of ILVs results in the formation of late endosomes (MVBs). Late endosomal markers include RAB7, RAB9, and mannose 6-phosphate receptors [30]. RAB5A, which is activated by caveolin-1, has been shown to be necessary for the formation of late endosomes [31]. RAB5A also regulates the secretion of exosomes. The fusion of MVBs with the plasma membrane results in the release of ILVs as exosomes [32] (Figure 1). These MVBs can fuse with lysosomes, leading to ILV destruction [32] (Figure 1). When fusing with lysosomes, the late endosomes lose RAB5A and acquire RAB7A. MVBs can also fuse with autophagosomes [32] (Figure 1).

**Figure 1 pharmaceutics-15-01465-f001:**
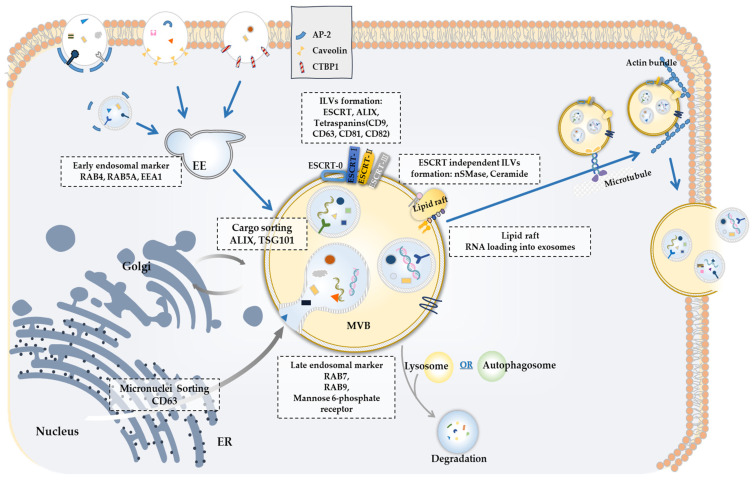
Overview of the exosome biogenesis. MVBs are derived from endocytosis [20]. Inward budding of the plasma membrane induces the formation of early endosomes [28,29]. Early endosomes sort various cargoes into ILVs to form MVBs [28,29]. After maturation, MVBs interact with Golgi, fuse with lysosomes to be degraded, fuse with plasma membrane to release exosomes (ILVs), or fuse with autophagosome to form amphisome [32]. Amphisome can fuse with lysosome to be degraded or fuse plasma membrane for exosome release [32]. MVBs containing exosome cargo are transported to the plasma membrane via microtubule, fuse with the cell surface and the ILVs are then secreted as exosomes [32].

MVB formation occurs in the endosomal sorting complex required for transport (ESCRT)-dependent and ESCRT-independent pathways. ESCRT is involved in the biogenesis of MVBs and ILVs [33] (Figure 1). ESCRT consists of four complexes: ESCRT-0, -I, -II, and -III. The components of ESCRT include apoptosis-linked gene 2-interacting protein X (ALIX), tumor susceptibility gene 101 (TSG101), and vacuolar protein sorting-associated protein 4 (VPS4) [34,35,36]. Clathrin, which is recruited by ESCRT-0, is necessary for sorting membrane cargo, recruiting lipid modifiers, and efficient ILV formation [37]. ALIX and TSGS101 recognize and classify ubiquitinated cargo [38,39]. TSG101 binds to annexin A 10 (ANXA10) and activates the mitogen-activated protein kinase (MAPK)/extracellular regulated kinase (ERK) pathway, which enhances the proliferation of papillary thyroid carcinoma cells [40]. There are high levels of ALIX and TS101 in the exosomes of prostate cancer cell lines [41]. Taken together, these previous reports suggest the important roles played by exosomal components in cancer initiation and progression. The ESCRT-0 complex sequesters ubiquitylated proteins in the endosomal membrane, while the ESCRT-I and -II complexes induce membrane deformation into buds with sequestered cargo, and ESCRT-III components induce vesicle scission [42]. ALIX recruits ESCRT-III and VPS4 to complete the formation of ILVs by sorting ubiquitylated proteins in the endosomal membrane [43].

Exosome formation requires RAB GTPases, tetraspanins (CD63, CD81, CD82, and CD9), and lipid-modifying enzymes such as sphingomyelinases [44]. RAB27A promotes the metastatic potential of melanoma cells by promoting the formation of pro-invasive exosomes [44] and regulates the secretion of exosomes in patients with gastric cancer [45]. The downregulation of RAB27A decreases the secretion of exosomes in gastric cancer cells and inhibits the peritoneal metastasis of gastric cancer cells without affecting the invasion potential of cancer cells [46]. ALIX binds to tetraspanins (CD9, CD63, and CD81) and promotes the sorting of the molecular cargo into late endosomes [47]. CD63 induces the generation of ILVs in late endosomes/MVBs [48].

Lipid rafts present in exosomal membranes are involved in sorting molecular cargo, membrane curvature, and vesicle budding. The ESCRT-independent pathway of ILV formation involves RNA loading into exosomes via the interactions of RNAs with exosomes and lipid rafts [49,50] (Figure 1). RNA binding proteins—such as heterogeneous nuclear ribonucleoprotein A2B1 (hnRNPA2B1), hnRNPK, Y box binding protein 1 (YBX1), and protein argonaute-2 (AGO2)—play important roles in miRNA sorting into exosomes [51,52,53]. The neutral sphingomyelinase nSMase converts sphingomyelin into ceramide, which plays a role in shaping membrane curvature and sorting cargo [54,55].

These reports suggest that proteins that are involved in the biogenesis of exosomes contribute to the pathogenesis of cancer. Therefore, it is necessary to achieve a complete understanding of the mechanisms of the formation and release of exosomes to develop anticancer therapy. Extensive studies elucidating the roles of exosomal contents will enhance the value of exosomes as targets for developing anticancer drugs.

## 3. Contents of Exosomes: Exosomal Proteins

Exosomes contain proteins, non-coding RNAs, DNA fragments, and lipids. Based on the fact that they show differential expression between normal and cancer cells, exosomal proteins can serve as prognostic and/or diagnostic markers of various diseases [56]. Exosomal proteins include: (1) membrane transport and fusion proteins, including annexin, RAB GTPases, and heat shock proteins (Hsp60, Hsp70, and Hsp90); (2) tetraspanins, including CD9, CD63, CD81, CD82, CD106, and intercellular adhesion molecule (ICAM); (3) MVB-related proteins, including ALIX and TSG101; and (4) cytoskeletal proteins, such as actin and myosin. To understand the roles played by these exosomal proteins in various life processes, it is useful to review the findings of prior relevant studies, which include the following: micronuclei containing gDNA and nuclear proteins interact with CD63 to be sorted into ILVs/exosomes [57] (Figure 1). Lysosome-associated membrane glycoprotein 2A (LAMP2A) regulates the loading of cytosolic proteins into exosomes [58]. ALIX is necessary for the formation and release of exosomes [59]. High levels of exosomal TSG101 are associated with aggressive phenotypes and the metastasis of colorectal cancers [60]. The downregulation of TSG101 decreases the secretion of WNT5B-containing exosomes from colorectal cancer cells [61]. Exosomal levels of receptor tyrosine kinases, such as platelet-derived growth factor receptor B, are decreased by chemotherapy in breast cancer patients [62], and the exosomes released from normal cells and cancer cells are quantitatively and qualitatively different [62]. These reports suggest that exosomal proteins play important roles in the biogenesis of exosomes, cancer initiation and progression, and the response to chemotherapy. Thus, these exosomal proteins have emerged as targets for developing anticancer drugs. Identifying the downstream targets of these proteins could also help us to achieve a better understanding of the pathogenesis of various diseases. Ultimately, it is necessary to understand the mechanisms of life processes mediated by these exosomal proteins.

## 4. Roles of Exosomes and Exosomal Proteins in Cellular Interactions and Antitumor Immune Responses

Exosomal proteins play important roles in various life processes [12,13,14,15,16,17,18,19] (Figure 2). It is known that tumorigenesis is closely associated with immune evasion [17]. Therefore, it is necessary to understand the roles played by exosomal proteins in tumor immune responses, which can be achieved by reviewing the findings of prior studies as follows: metastatic gastric cancer cells display a high expression of CD97 compared to non-malignant gastric cancer cells [63]. Exosomes mediate the CD97-promoted lymphatic metastasis of gastric cancer cells [63]. Anticancer drugs enhance the release of exosomes with heat shock proteins (HSP60, 70, and 90) from hepatocellular carcinoma cells (HCCs) [64]. Exosomal heat shock proteins promote natural killer (NK) cell activity to enhance antitumor responses in HCCs [64]. Exosomal B7H4 increases the expression of FOXP3, which induces immune evasion in glioblastoma cells [17]. Exosomal programmed death ligand-L1 (PD-L1) binds to programmed death-1 (PD-1) on T cells to inhibit cytolytic T lymphocyte (CTL) activity and promote melanoma metastasis [65]. Breast cancer cell-derived exosomal PD-L1 inhibits the response to immune checkpoint inhibitors by binding to PD-1 on T cells [66]. Macitentan (MAC), a Food and Drug Administration (FDA)-approved drug, decreases the binding of exosomal PD-L1 to PD-1, which enhances the CD8+ T-cell-mediated antitumor response [66]. Taken together, these reports indicate that exosomes can modulate antitumor immune responses. Thus, targeting exosomal PD-L1 could enhance the sensitivity of cancer cells to anti-PD-1 antibodies.

Tumorigenesis involves interactions among cancer cells, stromal cells, and immune cells [11]. It is therefore reasonable to expect that exosomes might also mediate these cellular interactions, as has been investigated by prior studies: M2-like macrophage-derived exosomes enhance the metastatic potential of non-small cell lung cancer cells (NSCLCs) through exosomal integrin αVβ3 [67]. Exosomal Hsp90α is necessary for the recruitment of stromal cells, such as keratinocytes, by various cancer cells [68]. This implies that exosomal Hsp90α plays a role in mediating cancer–stromal cell interactions. The exosomal membranous form of intercellular adhesion molecule-1 (ICAM-1) (mICAM-1) of prostate cancer cells inhibits the binding of leukocytes to endothelial cells activated by tumor necrosis factor-alpha [69]. Tumor-derived exosomal CD47 inhibits the phagocytic effects of macrophages by interacting with signal regulatory protein α (SIRPα) on phagocytes [70]. Exosomal CD9 of cancer-associated fibroblasts (CAF) suppresses the proliferation of melanoma cells [71].

The above reports indicate that exosomal proteins can mediate cellular interactions among cancer cells, immune cells, and stromal cells. Table 1 [17,46,61,63,64,65,66,67,68,69,70,71] and Figure 2 detail the roles played by exosomes and exosomal proteins in various life processes, such as cellular interactions, antitumor immune responses, and cancer cell proliferation. To better understand the mechanisms of cellular interactions mediated by these exosomal proteins, it is necessary to identify the targets of these exosomal proteins.

## 5. Exosomal Non-Coding RNAs as Regulators of Cancer Progression and Anticancer Drug Resistance

Exosomes affect recipient cells via surface molecules or molecular cargo [72]. Cellular interactions involve exosomal interactions with the plasma membrane of recipient cells, or by exosomal cargo release following the uptake of exosomes. [73]. Since exosomes contain non-coding RNAs [8,9,10], these non-coding RNAs may affect cancer progression and other life processes.

CAF-derived exosomal miR-181b-3p promotes colorectal cancer progression by decreasing the expression of the sorting nexin 2 (SNX2) gene [74]. CAF-derived exosomal miR-20a-5p decreases the expression of the LIM domain and actin binding 1 (LIMA1), which inhibits Wnt/β-catenin signaling in HCCs [75]. Tumor-derived exosomal miR-183-5p increases the number of PD-L1-expressing macrophages, which results in immune suppression and promotes intrahepatic cholangiocarcinoma (ICC) progression [76]. M2 macrophage-derived exosomal miR-193b-3p enhances the proliferation of pancreatic cancer cells by decreasing the expression of the tripartite motif containing 62 (TRIM62) [77]. Exosomal LINC00963 of lung cancer cells enhances the metastatic potential of lung cancer cells by decreasing the expression of seven in absentia homolog 1 (SIAH1) while increasing the zinc finger E-box-binding homeobox 1 (ZEB1) expression level [78]. Altogether, these reports suggest that targeting these miRNAs could suppress cancer initiation and progression by inhibiting cellular interactions.

Mesenchymal stem cell (MSC)-derived exosomal miR-1827 inhibits M2 macrophage polarization and the metastatic potential of CRC cells by decreasing the expression of succinate receptor 1 (SUCNR1) [79]. miR-27a-3p is decreased in the cells and tissues of patients with hepatic cancers. Exosomes derived from miR-27a-overexpressing MSCs suppress the progression of hepatic cancer by decreasing Golgi membrane protein 1 (GOLM1) expression [80]. Exosomal miR-10527-5p is decreased in both the plasma exosomes and tumor tissues of esophageal squamous cell carcinoma (ESCC) patients. Taken together, these studies indicate the roles played by exosomal miRNAs in cancer initiation and progression. Identifying the targets of these miRNAs would be helpful for understanding exosome-based tumorigenesis.

Since exosomal miRNAs regulate cancer cell proliferation, it is reasonable to expect that exosomal non-coding RNAs might modulate anticancer drug resistance. High levels of miR-374a-5p induce anticancer drug resistance in gastric cancer cells by binding to Neurod1 [81]. By contrast, the exosome-mediated delivery of miR-374a-5p inhibitor suppresses chemoresistance by increasing neuro D1 expression in gastric cancer cells [81]. The downregulation of SNHG11 induces apoptosis in bevacizumab-resistant CRC cells by targeting miR-1207-5p and decreasing the expression of ABCC1 [82]. Exosomal circSFMBT2 is highly expressed in the serum of docetaxel-resistant prostate cancer patients [83]. Exosomal circSFMBT2 enhances docetaxel-resistance by decreasing the expression of miR-136-5p while increasing the expression of tribbles homolog 1 [83]. miR-769-5p is enriched in the serum exosomes of cisplatin-resistant gastric cancer patients. Exosomes that have been loaded with miR-769-5p delivered into sensitive gastric cancer cells induce cisplatin resistance by decreasing the expression of caspase-9 and p53 [84].

miR-302c-5p inhibits cisplatin resistance and cancer stem cell-like properties in nasopharyngeal carcinoma (NPC) [85]. Exosomal miR-214 enhances the sensitivity of HCCs to oxaliplatin and sorafenib by decreasing the expression levels of P-gp and splicing factor 3B subunit 3 (SF3B3) [86]. Exosomal miR-204-5p enhances the sensitivity of CRC cells to 5-fluorouracil (5-FU) by inducing apoptosis [87].

These reports indicate the roles played by exosomal miRNAs in anticancer drug resistance. Table 2 shows the effects of exosomal non-coding RNAs on anticancer drug resistance and cancer progression. Targeting these exosomal miRNAs may provide clues for developing anticancer drugs. These reports suggest that exosomal non-coding RNAs can serve as targets for developing anticancer therapies. Mimics or inhibitors of these non-coding RNAs could be developed as anticancer drugs. In particular, mimics or inhibitors could be used in combination with chemotherapeutic drugs to overcome anticancer drug resistance. Table 2 [74,75,76,77,78,79,80,81,82,83,84,85,86,87,88] and Figure 3 show the roles played by exosomal non-coding RNAs in cancer progression and anticancer drug resistance.

## 6. Exosomal miRNAs as Diagnostic/Prognostic Markers

Since exosomes represent their cells of origin, contain various biomolecules, and are secreted into the bloodstream, they can be diagnostic and/or prognostic markers of cancers. It is first necessary to examine whether exosomal non-coding RNAs can be used as diagnostic markers. Plasma exosomal miR-320d, miR-4479, and miR-6763-5p are decreased in patients with epithelial cancers compared to healthy controls [4]. Increased levels of exosomes might represent a hallmark of malignant cancers, and they could serve as an indicator of clinical status [89]. Serum exosome-derived miRNAs (miR-122-5p, let-7d-5p, and miR-425-5p) can distinguish HCC patients from healthy donors [90]. The expression levels of exosomal miR-21, miR-155, miR-182, and miR-373 in the serum of breast cancer patients are higher than those in healthy controls [91]. Urinary exosomal lncRNAs can distinguish bladder cancer patients from healthy controls based on RNA sequencing [92]. Compared to healthy controls, salivary exosomal miR-486-5p is elevated and miR-10b-5p is reduced in oral and oropharyngeal squamous cell carcinoma [93]. These reports indicate that exosomal miRNAs can serve as diagnostic markers.

Gastric cancer patients with low serum exosomal miR-134 levels have shorter overall survival (OS) and relapse-free survival [94]. Exosomal LINC00265 and LINC00467 are increased in patients with acute myeloid leukemia (AML) who achieve complete remission after chemotherapy [95]. Exosomal miR-7-5p enhances the therapeutic efficacy of everolimus by inhibiting MAPK-interacting protein kinase (MNK)/eukaryotic initiation factor 4E (eIF4E) axis [88]. This implies that high levels of exosomal miR-7-5p could predict a favorable response to chemotherapy. Plasma exosomal lnc-SNAPC5-3:4 is significantly upregulated when anlotinib treatment is effective in patients with non-small cell lung cancers (NSCLCs) [96]. High levels of LINC00963 can predict a poor prognosis of patients with lung cancer [78]. LncRNAs—such as AL355353.1, AC011468.1, and AL354919.2—are upregulated in bladder cancer tissues [97]. High levels of AL354919.2 can predict poor prognosis in bladder cancer patients [97]. High levels of bile exosomal miR-200a-3p, miR-200c-3p, and serum exosomal miR-200c-3p can predict poor prognosis of patients in cholangiocarcinoma [98]. Low levels of circulating exosomal lncRNA-GC1 can predict a better prognosis from adjuvant chemotherapy than high levels of lncRNA-GC1 [99]. Collectively, these results indicate that exosomal non-coding RNAs can serve as prognostic markers.

Table 3 lists exosomal non-coding RNAs as diagnostic and prognostic markers. Table 4 presents clinical trials involving exosomal molecules (non-coding RNAs and proteins). These trials aimed to (1) validate exosomal non-coding RNAs and proteins as diagnostic and/or prognostic markers and (2) successfully isolate exosomes. Clinical reports have yet to be posted. It will be necessary to examine the functional roles of these non-coding RNAs in cancer initiation and/or progression.

## 7. Inhibitors Targeting Exosomes

Pools of exosomes are packed in MVBs and released into the extracellular space following the fusion of MVBs with the plasma membrane [100]. Since cancer cells secrete more exosomes than normal cells [2], inhibitors of exosome biogenesis/and or secretion can be developed as anticancer drugs. It is necessary to investigate the roles that these inhibitors play in cancer by reviewing the results of relevant studies.

Spiroepoxide, which is an inhibitor of nSMase, inhibits T-cell proliferation by decreasing exosome secretion from macrophages and dendritic cells (DCs) [101]. Spiroepoxide likely suppresses cancer cell proliferation by inhibiting cellular interactions within the tumor microenvironment. nSMase2 activity increases sEV secretion by modulating vacuolar H+-ATPase (V-ATPase) activity [102]. The inhibition of nSMase2 by GW4869 induces acidification and decreases the secretion of exosomes [102]. The inhibition of nSMase2 enhances the sensitivity of human epidermal growth factor receptor 2-overexpressing cells to trastuzumab [103]. Avicin G, which is an inhibitor of nSMase, suppresses ras signaling by depleting cholesterol contents in pancreatic cancer cells and NSCLCs [104]. Altogether, these reports suggest that the inhibition of exosomes biogenesis and/or secretion may suppress cancer cell proliferation.

GW4869 inhibits both the biosynthesis and release of exosomes in ovarian cancer cells [105]. GW4869 treatment reduces the number of exosomes released and inhibits the proliferation of paclitaxel-resistant prostate cancer cells [106]. GW4869 enhances the sensitivity of human myeloid leukemia cells to doxorubicin [107]. Exosomal PD-L1 induces resistance to paclitaxel by activating the STAT3/miR-21/phosphatase and tensin homolog 1 (PTEN)/Akt axis in esophageal cancer cells [108]. GW4869 reverses the paclitaxel resistance conferred by exosomal PD-L1 [108]. GW4869 inhibits both the release of exosomes from prostate cancer cells and M2 macrophage polarization [109]. It is known that M2 macrophages can enhance tumor initiation and progression [110]. Thus, GW4869 promotes tumor suppression by inhibiting M2 macrophage polarization. These reports highlight the roles played by exosomes in anticancer drug resistance as well as cancer initiation and progression.

RAB27B is involved in exosome biogenesis [111]. Moreover, the expression of RAB27B is increased in 5-FU-resistant hepatic cancer cells [112]. Thus, an RAB27B inhibitor could inhibit exosome biogenesis and suppress anticancer drug resistance. The knockout of RAB27A—a regulator of exosome biogenesis—enhances the sensitivity to chemo immunotherapy in B-cell lymphoma [113]. Tipifarnib, an inhibitor of RAB27A, enhances the sensitivity of various cancer cells to sunitinib by decreasing the number of exosomes released and the expression of PD-L1 [114]. These reports further suggest that targeting exosome biogenesis could provide valuable insights into overcoming anticancer drug resistance.

Exosomal PD-L1 induces immune evasion to promote cancer initiation and progression [115]. Reductions in exosomal PD-L1 by atorvastatin (ATO) enhance the efficacy of anti-PD-L1 blockade in breast cancer cells [116]. GW4869 enhances the effect of PD-L1 checkpoint blockade by stimulating CTL activity in a mouse model of melanoma [117]. Thus, exosome inhibitors can be used in combination with immune checkpoint inhibitors to achieve enhanced anticancer immune responses.

Either an increase in intracellular Ca2+ or the activation of protein kinase C is necessary for the secretion of exosomes in human red blood cells (RBCs) [118]. Thus, PKC inhibitors can reduce the number of exosomes secreted, which might suppress the tumor-promoting functions of exosomes.

The inhibition of rho-associated protein kinase (ROCK) decreases the number of exosomes that are secreted in response to rapamycin in primary articular chondrocytes [119]. Exosomes from the metastatic colon cancer cells induce endothelial hyperpermeability via the RhoA/ROCK pathway [120]. Exosomal LINC00161 promotes angiogenesis and the metastasis of hepatic cancer cells by activating ROCK signaling [121]. Thus, Y27362—an inhibitor of ROCK—could prevent exosome-mediated cellular interactions.

Exosomes from bone marrow-derived MSCs promote wound healing via MAPK signaling in human corneal epithelial cells [122]. U0126, which is an inhibitor of MAPK, prevents exosomes from enhancing the proliferation and migration of human corneal epithelial cells [122]. Thus, MAPK inhibitors could suppress cancer cell proliferation by exerting a functional block on exosomes.

Prior studies have identified many compounds that inhibit the biogenesis and/or release of exosomes. Extensive studies involving in vitro and in vivo models are required for developing these compounds as single or combination therapy drugs. It is therefore necessary to identify the mechanism of the anticancer effects exerted by these inhibitors. It is also necessary to identify the targets to understand the mechanisms of exosome-based cancer initiation and progression.

## 8. Exosomes as Carriers of Drug/siRNAs/miRNAs

### 8.1. Advantages of Exosomes as Carriers of Anticancer Molecules

Exosomes can deliver molecular cargo to target cells with high efficacy. The double-membrane structure of exosomes protects exosomal cargo [123]. The outer phospholipid membrane of exosomes improves their drug-targeting efficiency [123]. Exosomes exhibit efficient tumor enrichment effects, which are known as high permeability and retention effects (EPR) [124,125]. Exosomes also display good biocompatibility, low immunogenicity, stability, and low toxicity [126,127]. Tumor cell-derived exosomes display tumor-specific targeting [128]. All of these properties make exosomes promising delivery vehicles of anticancer molecules [129,130].

Exosomes play roles in every stage of cancer progression by mediating intercellular communication. Exosomes absorbed by recipient cells release their content, which regulates the fate of the recipient cells. The fact that exosomes can alter the immune system suggests that they can be exploited as a cell-free therapeutic approach for a variety of diseases. Since exosomal cargo can promote or inhibit tumor formation, exosomes can be developed as carriers of various anticancer drugs. Although miRNAs can inhibit tumor growth, they are easily degraded; using exosome as miRNAs carriers can solve this problem. Chemicals often display high toxicity, and proteins do not exert desired functions due to a lack of native conformation; using exosomes as carriers can solve these problems as well. Exosomes can deliver drugs to target sites over a great distance. Exosomes can cross cytoplasmic membranes and the blood–brain barrier [131,132]. Although natural exosomes cannot achieve the expected therapeutic effects, they can be modified to load various anticancer molecules.

### 8.2. Methods of Encapsulating Anticancer Molecules in Exosomes

Since modified exosomes have shown potential as delivery vehicles, it is necessary to discuss the advantages/disadvantages of different methods of encapsulating anticancer molecules. Several methods can be used to encapsulate drugs in exosomes: exogenous loading, endogenous loading, and fusion methods. Exogenous loading includes co-incubation, extrusion, electroporation, and sonication. Co-incubation often displays a low efficiency of drug-loading into exosomes. Drugs may cause cytotoxic effects during co-incubation with exosomes. Extrusion induces high cargo-loading efficiency; however, it can compromise the immune-privileged status of exosomes. Electroporation may induce high loading efficiency; however, it leads to the aggregation of molecular cargo. Sonication induces high loading efficiency; however, it impairs exosomal integrity and cargo aggregation. For each of these methods, it will be necessary to minimize any disadvantages to maximize the delivery potential of exosomes.

Endogenous loading packages specific molecules into exosomes by modifying donor cells. For example, the miR-145-5p transfection of MSCs leads to the production of MSC-derived exosomes with an accumulation of miR-145-5p [133]. The fusion method involves the fusion of exosomes with nano-liposomes. This method can improve drug loading efficacy while maintaining the functions of exosomes. These hybrid exosomes can be endocytosed by MSCs and subsequently release their cargo [134]. Future studies should aim to reduce any unnecessary immune reactions caused by these exosomes containing miRNAs. Figure 4 shows the advantages/disadvantages of each method of encapsulating anticancer molecules in exosomes.

### 8.3. Exosomes Loaded with Anticancer Drugs Inhibit Cancer Cell Proliferation

Exosomes can deliver cisplatin through clathrin-independent endocytosis and evade endosome trapping; in this way, exosomes can diffuse evenly in the cytosol [135]. This suggests that exosomes can deliver anticancer drugs to anticancer drug-resistant cancer cells. Exosomes lack immunogenicity and toxicity [126]. Therefore, exosomes are promising as carriers of anticancer drugs. It is necessary to examine the effects of modified exosomes on cancer cell proliferation and anticancer drug resistance.

**Figure 4 pharmaceutics-15-01465-f004:**
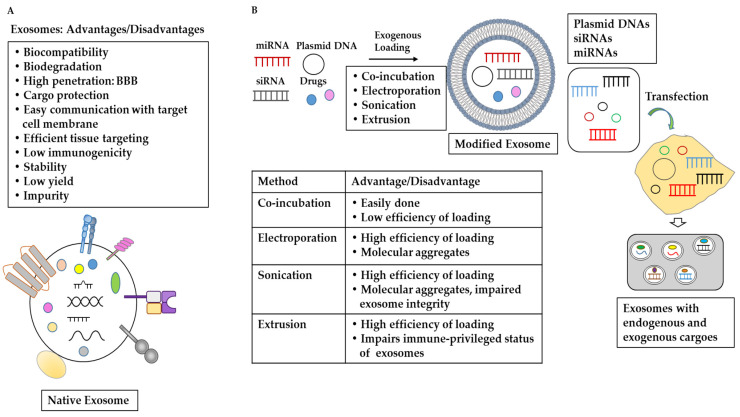
Exosomes as delivery vehicles. (**A**) Advantages and disadvantages of exosomes as delivery vehicles [123,124,125,126,127,128,129,130,131,132]. (**B**) Exogenous loading involves the loading of various cargoes, such as drugs, DNAs, ncRNAs, and proteins, into native exosomes. Methods for loading these cargoes include electroporation, sonication, co-incubation, and extrusion [136,137,138,139,140,141]. Endogenous loading involves the packaging of biomolecules via the modification of donor cells [133,134,142,143,144,145]. This modification of donor cells can be performed by means of the transfection of DNAs and various ncRNAs.

NK-cell-derived exosomes that have been loaded with paclitaxel via electroporation induce apoptotic effects in breast cancer cells by increasing the expression of caspase-3 and BAX [136]. Umbilical cord blood-derived M1 macrophage exosomes that have been loaded with cisplatin via electroporation show more potent effects against cisplatin-sensitive and -resistant ovarian cancer cells than free cisplatin [137]. Bone marrow mesenchymal stem cell (BMSC)-derived exosomes that have been loaded with doxorubicin show more potent effects against osteosarcoma than free doxorubicin [138]. These exosomal mimetics are prepared by the extrusion of BMSCs and show fewer side effects than free doxorubicin [138]. MSC-derived exosomes that have been loaded with doxorubicin via electroporation showed more potent effects than free doxorubicin in a mouse model of colon cancer [139]. Platelet exosomes that have been loaded with doxorubicin via electroporation decrease the proliferation of MDA-MB-231 cells by increasing the expression of BAX while decreasing the expression of BCL2 [140]. BMSC-derived exosomes that have been loaded with rifampicin via ultrasound inhibit the proliferation of osteosarcoma cells, induce G2/M arrest, and mitochondrial apoptosis [141]. In doing so, the exosomes activate dynamin-related protein 1 (Drp1) [141]. These reports show the therapeutic potential of modified exosomes. Identifying the targets of these modified exosomes may offer clues for understanding the mechanism of cancer cell proliferation.

### 8.4. Exosomes Loaded with miRNAs Inhibit Cancer Cell Proliferation

Since exosomal miRNAs play important regulatory roles in cancer initiation and progression, it is reasonable to expect that modified exosomes carrying miRNAs could regulate cancer cell proliferation. BMSC-derived exosomes coated with miR-22-3p via transfection inhibit the proliferation of human melanoma cells by suppressing EMT and targeting galectin 1 (LGALS1) [142]. Hepatic stellate cell-derived exosomes that have been loaded with miR-335-5p via transfection suppress both the proliferation of hepatic cancer cells and in vivo tumor growth [143]. NSCLC serum-derived exosomes that have been loaded with miR-126 via transfection suppress the proliferation of NSCLC and tumor growth by decreasing the expression of integrin α-6 [144]. HEK293T-derived exosomes that have been loaded with circ DIDO1 (Exo-circ DIDO1) via transfection suppress the progression of gastric cancer by decreasing the expression of miR-1307-3p but increasing the expression of suppressor of cytokine signaling 2 (SOCS2) [145]. Exo-circ DIDO1 displays higher inhibitory effects on the proliferation and invasion of gastric cancer cells than Exo-vector does [145]. Exo-circ DIDO1 does not induce obvious abnormalities or lesions in major tissues, such as the heart, liver, spleen, lungs, or kidneys [145].

Lung cancer-derived exosomes that have been loaded with miR-563 via co-incubation inhibit lung cancer cell proliferation, invasion, and induce apoptosis [146]. These exosomes show acceptable safety profiles and enrichment in lung cancer tissues [146]. HEK293T-derived exosomes that have been loaded with miR-34a via co-incubation inhibit the proliferation of oral squamous carcinoma cells by decreasing the expression of SATB homeobox 2 (SATB2) [147]. HEK293 T-derived exosomes that have been loaded with miR-317b-5b via co-incubation suppress the proliferation of lung cancer cells and induce apoptosis [148].

These reports indicate that exosomes can be employed as a delivery vehicle for non-coding RNAs such as miRNAs. For applications of exosomes in clinical trials, concerted efforts to improve pharmacokinetics are needed.

Since exosomal miRNAs have multiple targets [8,9,10], they might cause unwanted side effects. Reducing such side effects will be an important part of employing modified exosomes with miRNAs as anticancer therapies.

### 8.5. Exosomes Loaded with siRNAs Inhibit Cancer Cell Proliferation

The clinical applications of siRNA therapeutics have been limited by the immunogenicity of siRNA and the low efficiency of siRNA delivery to the target cells. Therefore, it is reasonable to expect that modified exosomes carrying siRNAs might offer better therapeutic options. HEK293T-derived exosomes that have been loaded with carboxypeptidase E (CPE)-small hairpin (shRNA) via transfection suppress the proliferation of HCCs by decreasing the expression of cyclin D1 and C-MYC [149]. Glioma cell-derived exosomes that have been loaded with PDGFRβ 9 siRNA via transfection suppress the proliferation of glioma cells by inhibiting the PI3K/Akt/EZH2 pathway [150]. Exo-PDGFRβ 9 siRNA shows no appreciable toxicity and high targeting activity [150]. NSCLC-derived exosomes that have been loaded with PD-L1 siRNA via electrostatic interaction induce the apoptosis in NSCLCs [151]. Exo-PD-L1siRNA shows enhanced downregulation of PD-L1 compared to Exo-control and also displays low toxicity and a high loading rate [151]. Immature DC-derived exosomes that have been loaded with B-cell lymphoma 6 (BCL6) siRNA via electroporation suppress the proliferation of diffuse large B-cell lymphoma (DLBCL) [152]. These exosomes deliver BCL6 siRNA to the tumor sites and do not induce noticeable toxicity [152]. Collectively, these results suggest that exosomes that have been loaded with siRNAs can be employed in anticancer therapies.

Tumor-derived exosomes can promote immune evasion by suppressing immune functions in cervical cancer cells [153]. Inducing intratumoral effector immune functions, in addition to suppressing immunosuppression, is the core of cancer immune therapy [153]. BMSC-derived exosomes loaded with galectin 9 siRNA via electroporation suppress the proliferation of pancreatic cancer cells by enhancing CTL activity and inhibiting Treg cell activity [154]. Exo-galectin 9 siRNA shows high tumor targeting efficiency and induces anti-tumorigenic M1 macrophage polarization [154]. Since exosomes that have been loaded with siRNA have potential in anticancer therapy, it is necessary to determine the clinical efficacy of exosomes.

## 9. Clinical Trials Involving Modified Exosomes

There are currently over 100 exosome-related clinical trials that have been registered at Clinicaltrials.gov. Although exosomes have shown substantial potential as anticancer therapy, many of these clinical trials have not been successful. Table 5 presents the clinical trials related to modified exosomes.

Exosomes derived from fibroblast-like MSCs were modified to carry siRNA or antisense oligonucleotide (ASO) targeting STAT3 (iExosiRNA-STAT3 or iExomASO-STAT3). Compared to scrambled siRNA control, siRNA-STAT3, or ASO-STAT3, iExosiRNA-STAT3 and iExomASO-STAT3 show improved STAT3-targeting efficiency [155]. iExosiRNA-STAT3 and iExomASO-STAT3 treatments decrease STAT3 levels and ECM deposition and also improve liver function in a mouse model of fibrosis [155]. MSC-derived exosomes carrying KrasG12D siRNA are being studied in a phase I clinical trial employing 28 participants (NCT03608631). This clinical trial aims to determine OS and progression-free survival (Table 5). It is necessary to examine whether exosomes carrying KrasG12D siRNA can induce antitumor responses.

Exosomes derived from IFN-γ-stimulated DCs (IFN-γ-Dex) were employed in a phase II clinical trial of 22 NSCLC patients. This clinical trial aimed to observe PFS at 4 mo. After the cessation of chemotherapy, fourteen patients (64%) showed stabilization, and eight patients (36%) experienced a partial response to platinum-based chemotherapy [156]. IFN-γ-Dex promotes NKp30-related NK cell functions [156]. The results of this clinical trial showed that IFN-γ-Dex could induce antitumor immunity by enhancing NK-cell activity. These reports suggest that modified exosomes could be developed as anticancer therapy.

A phase I clinical trial of exosomes containing curcumin is currently underway (NCT01294072) in 35 colon cancer patients. This trial measures the safety and the tolerability of curcumin and its effects on immune responses. The results of this clinical trial have yet to be posted. Curcumin is known to confer beneficial effects, such as anti-inflammation, antioxidant, and anticancer [157]. Nanoformulations of curcumin induced the osteogenic differentiation of MSCs [157]. Exosomes containing curcumin could likely enhance antitumor response by inducing CTL and/or NK cell activity.

Chimeric exosomal tumor vaccines prepared from antigen-presenting cells (APC)-tumor chimeric cells are being studied in a phase I clinical trial of nine bladder cancer patients (NCT05559177). This trial measures the clinical response rate, OS, and safety. The results of this clinical results have yet to be posted. A phase I trial showed the safety and feasibility of the vaccine. However, that phase I clinical trial did not monitor the induction of T-cell functions. Exosomes from tumor antigen loaded DCs were studied in a phase II clinical trial of 41 NSCLC patients (NCT01159288). This phase II clinical trial measured progression-free survival (PFS) in response to a combination of exosomes and chemotherapy. The results of this clinical trial have yet to be posted.

Although clinical trials involving exosomes have shown some success, such as low systemic toxicity, there are still challenges involved in exosome manufacturing, such as mass production and quality control (purity), particularly considering the size overlap between exosomes and other extracellular vesicles (e.g., MVs) or contamination (e.g., lipoprotein aggregates) during exosome production. To enhance the clinical value of exosome-based targeted delivery, a comprehensive understanding of the determinant attributes leading to the selective cell uptake of exosomes and factors influencing their intracellular disposition is needed.

## 10. Conclusions and Perspectives

Exosomes have shown potential as promising delivery agents because of their nontoxic and biodegradable characteristics as well as their ability to cross biological barriers, including the blood–brain barrier [158]. Exosomes also display long-term safety and effective tumor-targeting properties [159,160,161].

Since anticancer drugs affect signaling pathways that lie upstream of mitochondria [162], anticancer drugs that directly target mitochondria do not induce anticancer drug resistance. Therefore, it is reasonable to expect that targeting mitochondria could represent a promising strategy for treating various types of cancer. Polyphenolic compounds are known to regulate mitochondrial biogenesis [163]. Exosomes loaded with a polyphenolic compound may display improved anticancer effects without causing anticancer drug resistance. It is necessary to examine whether exosomes loaded with these mitochondria-targeting anticancer drugs could display improved pharmacokinetics and lower systemic toxicity. At present, there have yet to be any trials employing exosomes that have been loaded with mitochondria-targeting anticancer drugs.

The clinical application of exosome-based strategies in cancer patients has only been shown to lead to modest benefits. Many problems remain to be solved to advance the further application of exosomes. For example, the limited yield of exosomes could not satisfy the therapeutic application requirements in preclinical and clinical studies. Hybrid exosomes containing liposomes may increase exosome yield [134]. Naive exosomes show fast clearance, hepatic accumulations, and a lack of target-specific tropism. Therefore, the surface engineering of exosomes is a necessary step to reduce the non-specific adhesion of exosomes and improve their enrichment at the target site. Anchoring specific surface molecules on exosomes can increase the local concentration of exosomes at target sites, which reduces the toxicity and undesirable effects while maximizing therapeutic effects.

Exosomal content varies depending on the type of external stimuli, stage of disease, and the type of cancer. Exosomes have been shown to have different properties as well as different target specificity depending on cellular sources. For the clinical application of exosomes, it is necessary to understand how to select appropriate artificial exosomes to target different tumor types and how to deliver exosomes to recipient cells accurately and efficiently.

Many inhibitors targeting exosome biogenesis and/or release are currently being developed. Exosomes that have been loaded with these inhibitors could be employed as anticancer therapeutics. Identifying targets of these inhibitors will provide clues to better understand the mechanisms associated with cancer initiation and progression.

Unlike siRNAs, few clinical trials examining miRNAs have been conducted. Since miRNAs have multiple targets, it is probable that miR-inhibitors and miR-mimics could cause undesirable side effects. The use of exosomes with exogenously loaded miR-inhibitors or miR-mimics might reduce these side effects.

Exosomal proteins have been shown to play roles in exosome biogenesis, exosome release, and cellular communication. The structural identification of these proteins will provide clues to developing better anticancer drugs. Exosomes loaded with small molecules targeting these exosomal proteins could be developed as anticancer therapy.

In the future, we expect that more clinical trials will validate the clinical values of modified exosomes and exosomal contents. The problems associated with artificial exosomes will be solved to achieve more reliable and efficient anticancer therapeutics.

## Figures and Tables

**Figure 2 pharmaceutics-15-01465-f002:**
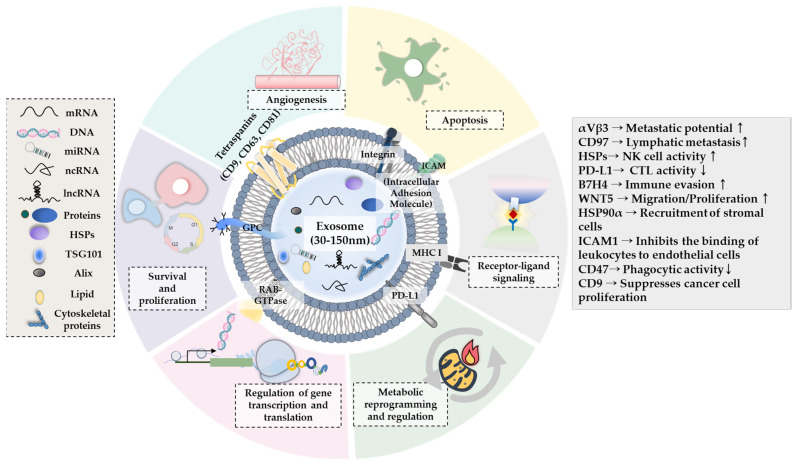
Roles of exosomes and exosomal proteins in life processes [17,46,61,63,64,65,66,67,68,69,70,71]. GPC, pre-glycoprotein polyprotein complex; ICAM, intercellular adhesion molecule; MHC, major histocompatibility complex; PD-L1, programmed death-ligand 1. ↓ denotes decreased expression/activity. ↑denotes increased expression/activity. → denotes direction of reaction.

**Figure 3 pharmaceutics-15-01465-f003:**
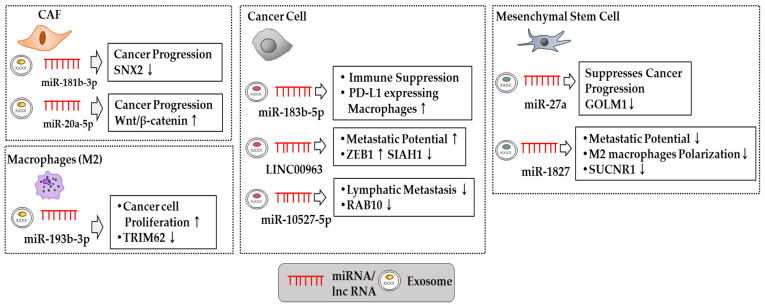
Diverse roles of exosomal non-coding RNAs. Exosomal miRNAs play important roles in cancer progression/metastasis [74,75,76,77,78,79,80], immune suppression [76], and M2 macrophages polarization [78]. Hollow arrows denote the direction of the reaction. ↓ denotes decreased expression/activity. ↑ denotes increased expression/activity. CAF denotes cancer-associated fibroblasts.

**Table 1 pharmaceutics-15-01465-t001:** Roles of exosomal proteins in cellular interactions. ↑ denotes increase in expression or activity.

Proteins	Origin of Exosomes	Effect	Ref.
B7H4	Glioblastoma cells	Enhances tumor growth by increasing the expression of FOXP3, which induces immune evasion.	[17]
RAB27A	Gastric cancer cells	Secretion of exosomes ↑ Peritoneal metastasis ↑	[46]
TSG101	Colorectal cancer cells	Enhances proliferation and migration of lung cancer cells via Wnt5b	[61]
CD97	Gastric cancer cells	Enhances proliferation and metastatic potential of gastric cancer cells	[63]
Heat shock proteins (Hsp60, Hsp70, Hsp90)	Hepatocellular carcinoma cells	Suppresses NK cell function	[64]
Integrins αV and β3	M2 macrophages	Enhances the metastatic potential of NSCLCs	[67]
HSP90	Breast cancer cells	Recruits stromal cells such as fibroblasts	[68]
ICAM-1 (membrane form)	Prostate cancer cells/Breast cancer cells	Suppresses adhesion of leukocytes to endothelial cells	[69]
CD47	HEK293T cells	Inhibits the phagocytic effect of macrophages	[70]
CD9	Cancer associated fibroblasts	Inhibits the proliferation of melanoma cells	[71]

**Table 2 pharmaceutics-15-01465-t002:** Effects of exosomal non-coding RNAs on anticancer drug resistance and cancer progression. ↓ denotes decrease in expression or activity. ↑ denotes increase in expression or activity.

Exosomal Cargo	Sources of Exosomes	Target	Effects	Mechanisms	Ref.
miR-181b-3p	Cancer associated fibroblasts	Colorectal cancer cells	Enhances progression of colorectal cancer	SNX2 ↓	[74]
miR-20a-5p	Cancer associated fibroblasts	Hepatic cancer cells	Enhances progression of hepatic cancer cells	LIMA1 ↓ Wnt/β-Catenin ↓	[75]
miR-183-5p	Intrahepatic cholangiocarcinoma	intrahepatic cholangiocarcinoma cells	Enhances progression of cholangiocarcinoma cells	Increases the number of PD-L1 expressing macrophages	[76]
miR-193b-3p	M2 macrophages	Pancreatic cancer cells	Enhances proliferation of cancer cells	TRIM62 ↓	[77]
LINC00963	Lung adenocarcinoma cells	Lung adenocarcinoma cells	Induces M2 polarization of macrophages, enhances metastasis of lung adenocarcinoma cells.	SIAH1 ↓ ZEB1 ↑	[78]
miR-10527-5p	Plasma and serum of esophageal squamous cell carcinoma (ESCC) patients	esophageal squamous cell carcinoma	Inhibits lymphatic metastasis of ESCC	Rab10 ↓	[9]
miR-1827	Human umbilical cord mesenchymal stem cells	Colorectal cancer cells	Suppresses metastatic potential	SUCNR1 ↓	[79]
miR-27a-3p	Mesenchymal stem cells	Hepatic cancer cells	Inhibits progression of hepatic cancer	GOLM1 ↓	[80]
circ_0063526	CDDP-resistant Gastric cancer cells	CDDP-sensitive gastric cancer cells	Enhances CDDP resistance	miR-449 ↓ SHMT2 ↑	[16]
miR-374a-5p	Serum of gastric cancer patients	Gastric cancer cells	Enhances resistance to oxaliplatin	NeuroD1 ↓	[81]
Lnc RNA SNHG11	Bevacizumab-resistant Colorectal cancer cells	Colorectal cancer cells	Enhances Bevacizumab resistance	MiR-1207-5p ↓ ABCC1 ↑	[82]
circSFMBT2	Docetaxel-resistant prostate cancer cells	Prostate cancer cells	Enhances resistance to doxorubicin	miR-136-5p ↓ TRIB1 ↑	[83]
miR-769-5p	CDDP-resistant Gastric cancer cells	CDDP-sensitive gastric cancer cells	Enhances CDDP resistance	CASP9 ↓ P53 degradation by ubiquitination	[84]
miR-214	Human cerebral endothelial cells	HCC	Enhances the sensitivity to oxaliplatin and sorafenib	P-gp ↓ Splicing factor 3B subunit 3 (SF3B3) ↓	[86]
miR-204-5p	HEK293T cells	Colorectal cancer cells	Enhances the sensitivity to 5-FU	RAB22A ↓ Bcl-2 ↓	[87]
miR-7-5p	NSCLC	NSCLC	Enhances the sensitivity to Everolimus	MNK/eIF4E ↓ mTOR ↓	[88]

**Table 3 pharmaceutics-15-01465-t003:** Exosomal non-coding RNAs as diagnostic and prognostic markers.

miRNAs/Lnc RNAs	Cancer Types/Sources	Diagnosis/Prognosis	Expression	Ref.
miR-320d, miR-4479, and miR-6763-5p	Epithelial ovarian cancer/Plasma	Diagnosis	Downregulated in cancer patients	[4]
miR-122-5p, let-7d-5p, and miR-425-5p	Hepatocellular carcinoma/Serum	Diagnosis	Increased in cancer patients	[90]
miR-21, miR-155, miR-182, miR-373, and miR-126	Breast cancer/Serum	Diagnosis	Increased in cancer patients	[91]
MKLN1-AS, TALAM1, TTN-AS1 and UCA1	Bladder cancer/Urine	Diagnosis	Increased in cancer patients	[92]
miR-10b-5p and miR-486-5p	Oral and oropharyngeal cancer/Saliva	Diagnosis	miR-486-5p is elevated and miR-10b-5p is decreased	[93]
miR-134	Gastric cancer/Serum	Diagnosis	Downregulated in cancer patients	[94]
LINC00265, LINC00467, and UCA1	Acute myeloid leukemia/Plasma	Diagnosis	Downregulated in cancer patients	[95]
lnc-SNAPC5-3:4	NSCLC/Plasma	Prognosis	High level can predict favorable response to anlotinib	[96]
lncRNA AL355353.1, AC011468.1, and AL354919.2	Bladder cancer/Urine	Prognosis	High level can predict unfavorable prognosis	[97]
miR-200c-3p	Cholangiocarcinoma/Serum	Prognosis	High level can predict unfavorable prognosis	[98]
lncRNA-GC1	Gastric cancer	Prognosis	Low level can predict favorable response to 5-FU treatment	[99]

**Table 4 pharmaceutics-15-01465-t004:** Clinical trials involving exosomes.

Title	Status	Condition or Disease	Prospective Outcome Measures	Dates	ID/Purpose
Serum Exosomal Long Noncoding RNAs as Potential Biomarkers for Lung Cancer Diagnosis	Completed	Lung cancer	Enrollment number: 1000 The expression levels of serum exosome long non-coding RNA and tumor biomarkers such as CEA, NSE, SCC, and CYFR2A-1	Start: 1 January 2017 Completion: 31 December 2020	NCT03830619/Diagnosis
Interrogation of Exosome-mediated Intercellular Signaling in Patients with Pancreatic Cancer	Recruiting	Pancreatic cancer	Enrollment number: 111 (1) Successful isolation of exosomes (2) Evaluation of the liver microenvironment for alterations in cellular infiltrates and ECM	Start: March 2015 Completion: March 2024	NCT02393703/Diagnosis
A Pilot Study of Circulating Exosome RNA as Diagnostic and Prognostic Markers in Lung Metastases of Primary High-Grade Osteosarcoma	Active, not recruiting	Osteosarcoma, lung metastases	Enrollment number: 90 The correlation between circulating exosome RNA mutation levels and 3-year disease-free survival (DFS), progression-free survival (PFS), lung metastases	Start: 1 May 2017 Completion: 19 September 2022	NCT03108677/Diagnosis and prognosis
Clinical Validation of a Urinary Exosome Gene Signature in Men Presenting for Suspicion of Prostate Cancer	Completed	Prostate cancer	Enrollment number: 2000 Correlate an exosome gene expression signature with the presence or absence of high-grade prostate cancer in the prostate needle biopsy.	Start: May 2014 Completion: April 2015	NCT02702856/Diagnosis
Use of Circulating Exosomal lncRNA-GC1 as Blood Biomarker for Early Detection and Monitoring Gastric cancer	Recruiting	Gastric cancer	Enrollment number: 700 Detection of levels of circulating exosomal lncRNA-GC1 by RT-PCR	Start: 1 May 2022 Completion: December 2023	NCT05397548/Diagnosis
A Prospective, Multicenter Cohort Study of Urinary Exosome lncRNAs for Preoperative Diagnosis of Lymphatic Metastasis in Patients with Bladder Cancer	Not yet recruiting	Bladder cancer	Enrollment number: 74 (1) Analyses of number and location of lymph node metastases based on pathological diagnosis (2) The rate of recurrence will be detected after radical cystectomy by postoperative follow-up visit	Start: 1 June 2022 Completion: 1 August 2025	NCT05270174/Diagnosis
A Prospective Study of Predicting Prognosis and Recurrence of Thyroid Cancer Via New Biomarkers, Urinary Exosomal Thyroglobulin and Galectin-3	Active, not recruiting	Thyroid cancer	Enrollment number: 74 The correlation of outcome (including recurrence, lymph nodes metastasis) together with unknown/fresh biomarkers: Thyroglobulin, galectin-3, Calprotectin A8, Calprotectin A9, TKT, Annexin II, Afamin, Keratin 8, Keratin 9, Angiopoietin-1 and TIMP	Start: 3 August 2018 Completion: 31 July 2023	NCT03488134/Prognosis

**Table 5 pharmaceutics-15-01465-t005:** Clinical trial involving modified exosomes.

Title	Arms/Interventions	Study Design	Types of Cancers	Phase	Study Dates	NCT Number
Phase I Study of Mesenchymal Stromal Cells-Derived Exosomes with KrasG12D siRNA for Metastatic Pancreas Cancer Patients Harboring KrasG12D Mutation	Participants receive mesenchymal stromal cells-derived exosomes with KrasG12D siRNA IV over 15–20 min on days 1, 4, and treatment repeats every 14 days for up to 3 courses	Outcome measures: (1) Maximum Tolerated Dose (2) Overall survival (3) Progression free survival	Metastatic Pancreatic Adenocarcinoma	Phase 1	Start: 27 January 2021 Completion: 31 March 2023	NCT03608631
Phase I Clinical Trial Investigating the Ability of Plant Exosomes to Deliver Curcumin to Normal and Malignant Colon Tissue	Arm1: curcumin alone Arm 2: curcumin with plant exosome Arm 3: no treatment	Outcome measures: (1) Concentration of curcumin in normal and cancerous tissue (2) safety and tolerability of curcumin with plant exosomes (3) effects of curcumin with plant exosomes on immune responses	Colon cancer	Phase 1	Start: January 2011 Completion: January 2022	NCT01294072
An Open, Dose-escalation Clinical Study of Chimeric Exosomal Tumor Vaccines for Recurrent or Metastatic Bladder Cancer	APC-tumor chimeric cells were constructed and stimulated with immune stimulator. Chimeric exosomal vaccines were extracted from cell supernatants.	Type: Interventional Outcome measures: (1) Clinical response rate (2) Overall survival (3) Safety	Bladder cancer	Early phase 1	Start: 1 September 2011 Completion: June 2023	NCT05559177
Phase II Trial of a Vaccination with Tumor Antigen-loaded Dendritic Cell-derived Exosomes on Patients with Unresectable Non-Small Cell Lung Cancer Responding to Induction Chemotherapy	Metronomic dosage of Cyclophosphamide during 3 weeks (50 mg/day orally) Induction immunotherapy Intradermal injections of Dex once a week during 4 consecutive weeks. Continuation Immunotherapy: Intradermal injections of Dex every two weeks during 6 weeks.	Purpose: Treatment Type: Interventional Outcome measures: Progression free survival	Non-small-cell lung cancer	Phase 2	Start: 19 May 2010 Completion: 19 December 2015	NCT01159288

## Data Availability

Not applicable.

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
