# Peer review of "Exosomes: Diagnostic and Therapeutic Implications in Cancer"

_pharmaceutics, 2023, doi:10.3390/pharmaceutics15051465_

Round 1

Reviewer 1 Report

Dear Authors the description of exosomes in cancer diagnostic and therapy is interesting and supported by a robust and recent bibliography. Figure and tables are well made and rich of content.

Unfortunately the text must necessarily be revised by a native English speaker.

Several corrections must be made to the text as following reported:

Line 8, please remove the double in

Line 11, please replace RNAs with nucleic acids

Line 13, 19, 20 please rewrite sentences without repeating continuously “this review”

Line 28, 30, 31, 32, 34, 35, 37, 38 please rewrite sentences without repeating continuously “exosomes”

Figure 1 Legend:

Line 47, please replace for with to form

Line 48, please replace golgi qith Golgi

Line 48, 49, 50 please rewrite sentences without repeating continuously MVB

Line 56, please replace are originated with originate

Line 64, please replace cytoplasmic with plasma membrane

Line 101 and 102, please replace promoted and regulated with promotes and regulates

Lines 101-105 please rewrite sentences, without repetition, making speech smoother, easy to read

Line 109, please transform the sentence into active form

Line 111-115 please rewrite sentences without repeating continuously “lipid rafts”

 Line 149 please replace shave with have

Line 151 please remove space

Lines 153-176 please rewrite sentences with verbs to the present tense

Lines 195-231 please rewrite sentences with verbs to the present tense

Lines 264-287, please attention to the use of tenses of verbs. Here, too, continue to use the Simple present/ present tense.

Plese, continuing along the rest of the text, pay attention to the use of verb tenses and do not repeat the subject of the sentences too many times. Please combine phrases with the same subject.

Line 346, please remove space

Line 402 please remove NK-Exos because it is mentioned only once in the text

Line 466 plese insert the Reference [Zhou W et al., 2021] with the number 154

Line 523, please write in full BBB

Line 556 please replace theses with these

Please, check the References, and standardize them according to the journal’s requests.

Author Response

Dear Sir

Thanks for excellent suggestions. I made changes to accommodate suggestions from you. I hope that changes I made are fine. I sought professional help for English problems. I send English certificate.

Sincerely yours

Jeoung Dooil, Ph.D.

Professor of Biochemistry

Kangwon National University

Chuncheon 24341, Korea 

  • Line 8, please remove the double in

Ans. I took care of it. Thanks. Please take look at new manuscript (line 8).

  • Line 11, please replace RNAs with nucleic acids

Ans. I took care of it. Please take look at new manuscript (lines 12-13)

  • Line 13, 19, 20 please rewrite sentences without repeating continuously “this review”

Ans. I rewrote sentences. Please take look at new abstract.  

  • Line 28, 30, 31, 32, 34, 35, 37, 38 please rewrite sentences without repeating continuously “exosomes”

Ans. I rewrote sentences as you suggested. In this revision, I try not to repeat continuously with exosomes. 

Figure 1 Legend:

  • Line 47, please replace for with to form

Ans. I replace it (Line 48). Thanks.

  • Line 48, please replace golgi qith Golgi

Ans. I replace it (Line 48). Thanks.

  • Line 48, 49, 50 please rewrite sentences without repeating continuously MVB

Ans. I rewrite sentences. Thanks. Please take look at new figure legend 1.

  • Line 56, please replace are originated with originate

Ans. I change it. Please take look at line 56.  

  • Line 64, please replace cytoplasmic with plasma membrane

Ans. I change it. Please take look at line 66.

  • Line 101 and 102, please replace promoted and regulated with promotes and regulates

Ans. I change it. Please take look at lines 104-106.

  • Lines 101-105 please rewrite sentences, without repetition, making speech smoother, easy to read

Ans. Thanks. I rewrote sentences to make them more readable. Please take look at lines 104-109.   

  • Line 109, please transform the sentence into active form

Ans. I change the sentence into active form. Please take look at lines 113-114. 

  • Line 111-115 please rewrite sentences without repeating continuously “lipid rafts”

Ans. I change sentences without repeating ″lipid rafts”. Please take look at lines 113-120.  

  • Line 149 please replace shave with have

Ans. I change it into: Exosomes play important roles in various life processes [12-19]. Please take look at line 154.  

  • Line 151 please remove space

Ans. I took care of it. I remove this sentence: Since exosomes mediate cellular communications, exosomes play diverse roles in various life processes.  

  • Lines 153-176 please rewrite sentences with verbs to the present tense

Ans. Thanks. I rewrite sentences as you suggested. Please take look at lines 157-184.

  • Lines 195-231 please rewrite sentences with verbs to the present tense

Ans. Thanks. I rewrite sentences as you suggested. Please take look at lines 203-245.

  • Lines 264-287, please attention to the use of tenses of verbs. Here, too, continue to use the Simple present/ present tense.

Ans. I try to use the simple present tense as you suggested. Please take look at lines 263-287. I rearrange order of references in section 6 to make this manuscript more readable.  

  • Plese, continuing along the rest of the text, pay attention to the use of verb tenses and do not repeat the subject of the sentences too many times. Please combine phrases with the same subject.

Ans. Thanks. I agree. In this revision, I pay attention to the use of verb sentence. I remove unnecessary and repetitive sentences. In this revision, I try to combine phrases with same subject. I rearrange order of references. I try to make this manuscript more readable.      

  • Line 346, please remove space

Ans. Thanks. I remove space. Please take look at line 361-362.

  • Line 402 please remove NK-Exos because it is mentioned only once in the text

Ans. Thanks. I remove NK-Exos. Please take look at line 422-423.

  • Line 466 plese insert the Reference [Zhou W et al., 2021] with the number 154

Ans. Thanks. [Zhou W et al., 2021] is changed into [154]. Please take look at lines 488-489.

  • Line 523, please write in full BBB

Ans. I write in full BBB. Please take look line 542. 

  • Line 556 please replace theses with these

Ans. Please take look at line 582.

  • Please, check the References, and standardize them according to the journal’s requests.

Ans. Thanks. I do so as you suggested. Please take look new references

Reviewer 2 Report

This review summarized the role of exosomes in the diagnosis and treatment of cancer through the delivery of molecular cargo such as anticancer drugs, small interfering RNAs (siRNAs) and microRNAs (miRNAs). In addition, This review also discussed potential value of modified exosomes as carriers of drugs and provided current progress in the development of exosome inhibitors and clinical trials involving exosomes.

The topic selection of this manuscript is hot and the content is quite sufficient. In my opinion, the following points need to be clarified:

1. There are a lot of writing problems in the manuscript, which require the author to read through carefully, such as two "in" in the line 8, there is an extra "~" in line 29, line 149 has an extra "s" in "have", line 161 "PD-L1 bound to PD-L1", Line 466 [Zhou W et al., 2021] and so on.

2. The content of the manuscript is already quite sufficient, and I think it can be modified logically to facilitate the reader's better understanding. For example, There has been some accumulation of summaries of roles and diagnostic markers of exosomal proteins, non-coding RNAs, miRNAs. I suggest summarizing in terms of disease classification, such as in Metastatic Pancreas, Metastatic Bladder Cancer or Lung Cancer, which exosome markers can be directly indicated and used in the process of diagnosis and treatment. The content summarized below headings 4- 8 can be logically classified by disease.

3. In the table, similar diseases can be combined. For example, the roles of exosomes in Bladder cancer and Gastric cancer in Tablec3 can be combined.

Author Response

Dear Sir

Thanks for excellent suggestions. I made changes to accommodate suggestions from you. I hope that changes I made are fine. I sought professional help for English problems. I send English certificate.

Sincerely yours

Jeoung Dooil, Ph.D.

Professor of Biochemistry

Kangwon National University

Chuncheon 24341, Korea 

  • Q1. There are a lot of writing problems in the manuscript, which require the author to read through carefully, such as two "in" in the line 8, there is an extra "~" in line 29, line 149 has an extra "s" in "have", line 161 "PD-L1 bound to PD-L1", Line 466 [Zhou W et al., 2021] and so on.

Ans. Thanks. I made changes as you suggested. Line 8, line 32, line 154, lines 164-165, lines 488-489. In this revision, I try to make this manuscript more readable. In this revision, I remove unnecessary and repetitive sentences. I change order of references to make this manuscript more readable. I sought professional help for English problems. I send English certificate.

  • Q2. The content of the manuscript is already quite sufficient, and I think it can be modified logically to facilitate the reader's better understanding. For example, There has been some accumulation of summaries of roles and diagnostic markers of exosomal proteins, non-coding RNAs, miRNAs. I suggest summarizing in terms of disease classification, such as in Metastatic Pancreas, Metastatic Bladder Cancer or Lung Cancer, which exosome markers can be directly indicated and used in the process of diagnosis and treatment. The content summarized below headings 4- 8 can be logically classified by disease.

Ans. I understand your concern. However, I do not know if there are lung cancer or gastric cancer specific exosomal non-coding RNAs or proteins. I thought that it was difficult to classify exosomal markers based on cancer specificity. In this revision, I change tables to make them more readable. I describe roles of exosomal proteins in cancer cell proliferation/inhibition and cellular interactions (table 1). I describe roles of exosomal non-coding RNAs in cancer cell proliferation and anticancer drug resistance (table 2). I try to classify exosomal non-coding RNAs into diagnostic and prognostic markers (table 3). I also rearrange order of references to make this manuscript more readable. I know that this might not be the best classification. I hope that I do not cause some confusion or trouble.                

  • Q3. In the table, similar diseases can be combined. For example, the roles of exosomes in Bladder cancer and Gastric cancer in Tablec3 can be combined.

Ans. I understand your concern. Many exosomal non-coding RNAs employed as diagnostic markers have not been confirmed as prognostic markers. Therefore, I classify exosomal non-coding RNAs into diagnostic and prognostic markers. Of course, it is very possible that exosomal non-coding RNAs can be both diagnostic and prognostic markers. In this revision, I change table 3 to classify exosomal non-coding RNAs into diagnostic and prognostic markers. Please take look at new table 3. I 3 hope that this does not confuse you

Reviewer 3 Report

This review manuscript entitled "Exosomes: Diagnostic and Therapeutic Implications in Cancer" summarizes recent advances in developing exosomes as drug delivery platforms. The authors also present clinical trials involving exosomes.

This manuscript was well-thought but not well-executed rigorously, unfortunately, it has to be rejected due to the following reasons:

  1. Line 8: "… roles in in tumor initiation…" seems duplicate "in".
  2. Line 11: Exosomes not only contain proteins, lipids, and RNAs, but also contain DNA fragments. (Line 35 and line 123 were also not consistent.)
  3. Line 71: "Late endosomal markers include RAB7, RAB9, and mannose 6-phosphate receptors" should include citations.
  4. Figure 2-4: The font size of the text is too small.
  5. Line 151: Extra space before "Since exosomes …"
  6. Table 4: The arrangement of table 4 was awful since the information was not clearly classified, please refer to the tables from Perocheau et al (DOI: 10.1111/bph.1543).
  7. Table 5: Same issue as Table 4.
  8. The logical structure of the manuscript should be reconsidered, please refer Chen et al (DOI: 0.1007/s00018-019-03233-y). 
  9. Although authors may not be English native speakers, there are significant flaws in language and grammar.
  10. The technology of drug loading in exosomes should be a separate section.
  11. Take home message was poor: most of the manuscript was a list of literature, but lack of authors' views and insights.

Author Response

Dear Sir

Thanks for excellent suggestions. I made changes to accommodate suggestions from you. I hope that changes I made are fine. I remove unnecessary and repetitive sentences. I change order of references to make this manuscript more readable. I also make changes to make tables more readable. I sought professional help for English problems. I send English certificate.

Sincerely yours

Jeoung Dooil, Ph.D.

Professor of Biochemistry

Kangwon National University

Chuncheon 24341, Korea 

  • Q1. Line 8: "… roles in in tumor initiation…" seems duplicate "in".

Ans. I took care of it. Please take look at new manuscript.

  • Q2. Line 11: Exosomes not only contain proteins, lipids, and RNAs, but also contain DNA fragments. (Line 35 and line 123 were also not consistent.)

Ans. I took care of it. Please take look at lines 12-13, 38-39, and line 127.

  • Q3. Line 71: "Late endosomal markers include RAB7, RAB9, and mannose 6-phosphate receptors" should include citations.

Ans. I include citation [30]. Please take look at lines 75-76.

  • Q4. Figure 2-4: The font size of the text is too small.

Ans. Thanks. I took care of it. Please take look at new figure legends.

  • Q5. Line 151: Extra space before "Since exosomes …"

Ans. I took care of it. I delete "Since exosomes …"  I think this sentence is repetitive.

  • Q6. Table 4: The arrangement of table 4 was awful since the information was not clearly classified, please refer to the tables from Perocheau et al (DOI: 10.1111/bph.1543).

Ans. Thanks. I agree. I could not find DOI: 10.1111/bph.1543 in PubMed. Table 4 shows clinical trials involving exosomes. These clinical trials aim at validation of exosomal contents as diagnostic or prognostic markers. In this revision, I try to simplify table 4. In table 4, I first show clinical trials for validation of exosomes as diagnostic markers, and then those for validation of exosomes as prognostic markers.

  • Q7. Table 5: Same issue as Table 4.

Ans. Table 5 shows clinical trials involving modified exosomes. In this revision, I first try to simplify table 5 to make it more readable. Modified exosomes contain siRNAs and anticancer drug. I first wanted to show feasibility of exosomes loaded with siRNA or anticancer drug as anticancer therapy.

Since there have not been many clinical trials involving modified exosomes, I add clinical trials of immune cell-derived exosomes in cancer patients in Table 5. In this revision, I try to describe clinical trials in more detail in the manuscript.

  • Q8. The logical structure of the manuscript should be reconsidered, please refer Chen et al (DOI: 0.1007/s00018-019-03233-y). 

Ans. In this revision, I try to remove unnecessary and repetitive sentences. I divide section 8 into 5 subsections to make this manuscript more readable. I change abstract to make this manuscript more readable. In some sections, I rearrange order of references to make this manuscript more readable. 

  • Q9. Although authors may not be English native speakers, there are significant flaws in language and grammar.

Ans. In this revision, I sought professional for English problems. I send English certificate.

  • Q10. The technology of drug loading in exosomes should be a separate section.

Ans. Thanks. I agree. In this revision, I make separate section (section 8.2).

  • Q11. Take home message was poor: most of the manuscript was a list of literature, but lack of authors' views and insights.

Ans. I understand your concern. I agree. In this revision, I try to add my views and insights throughout the manuscript. I try to convey message more clearly. In this revision, I try to make this manuscript more informative. Thanks for your concern

Reviewer 4 Report

The review manuscript entitled " Exosomes-Diagnostic and therapeutic implications" is an interesting article describing the therapeutic application and mechanism of their secretion and the significance of various markers. The article is focused mostly towards therapeutic application and some focus on diagnosis. I have few comments on this manuscript version. 

1. The review covers several new aspects like clinical trials using modified exosomes, which is informative. At the same time some of the tables are very elaborate, which can be shortened. For example table 4, describes the study design in detail, which is interesting but certain information can be precise or can be moved to text.

2. The references should be properly formatted. It appears as though some of the reference numbers were deleted. 

Author Response

Dear Sir

Thanks for excellent suggestions. I made changes to accommodate suggestions from you. I hope that changes I made are fine. I sought professional help for English problems. I send English certificate.

Sincerely yours

Jeoung Dooil, Ph.D.

Professor of Biochemistry

Kangwon National University

Chuncheon 24341, Korea 

  • Q1. The review covers several new aspects like clinical trials using modified exosomes, which is informative. At the same time some of the tables are very elaborate, which can be shortened. For example table 4, describes the study design in detail, which is interesting but certain information can be precise or can be moved to text.

Ans. Thanks for your concern. I agree. In this revision, I simplify tables in general. Please take look at new tables. In this revision, I try to make tables more readable.          

  • Q2. The references should be properly formatted. It appears as though some of the reference numbers were deleted. 

Ans. I understand your concern. In this revision, I delete some unnecessary references. I change order of references to make this manuscript more readable. In this revision, I believe I properly format references

Reviewer 5 Report

This manuscript deals with "Exosomes: Diagnostic and therapeutic implications in cancer". I suggest a minor correction and require a detailed clarification. Correction to be addressed by the authors as follows: The abstract is not well organized, where the sentences are incomplete and no continuity is there. It would be feasible, if include the significance of the current study in the abstract. A brief description of how the authors selected information from the literature in the databases, as well as what time period they searched for, is missing.

Authors should justify and expand the information on the advantages of this study for biomedical applications. Authors should specify the main experimental conditions used on the evidences from the literature. Where they briefly describe the most important data reported in the literature in a homogeneous manner and sequence reinforcing the relevance of of this approach in targeting of neurobehavioral dysfunctions.

Authors should discuss whether the use of these method represents a solid alternative to existing methods. Also please discuss about the role of mitochondria targeting.

Please add below studies to your manuscript in discussion section using below manuscripts:

DOI: 10.1155/2021/4946711

DOI: 10.1155/2021/1520052

Conclusions should reaffirm the fundamental contribution of this paper.

Author Response

Dear Sir

Thanks for excellent suggestions. I made changes to accommodate suggestions from you. I hope that changes I made are fine. I sought professional help for English problems. I send English certificate.

Sincerely yours

Jeoung Dooil, Ph.D.

Professor of Biochemistry

Kangwon National University

Chuncheon 24341, Korea 

  • Q1. This manuscript deals with "Exosomes: Diagnostic and therapeutic implications in cancer". I suggest a minor correction and require a detailed clarification. Correction to be addressed by the authors as follows: The abstract is not well organized, where the sentences are incomplete and no continuity is there. It would be feasible, if include the significance of the current study in the abstract. A brief description of how the authors selected information from the literature in the databases, as well as what time period they searched for, is missing.

Ans. I agree. Thanks for your concern. In this revision, I change abstract to accommodate your suggestions. I try to change abstract to make this manuscript more readable. Please take look at new abstract.     

  • Q2. Authors should justify and expand the information on the advantages of this study for biomedical applications. Authors should specify the main experimental conditions used on the evidences from the literature. Where they briefly describe the most important data reported in the literature in a homogeneous manner and sequence reinforcing the relevance of of this approach in targeting of neurobehavioral dysfunctions.

Ans. In this study, I mention the advantages of exosomes as delivery vehicles. I mention various methods of encapsulating anticancer molecules in exosomes. I also mention advantages/disadvantages of each method. In this revision, I try to be more specific and include more information on each reference. In this revision, I try to add my views and insights.

  • Q3. Authors should discuss whether the use of these method represents a solid alternative to existing methods. Also please discuss about the role of mitochondria targeting.

Please add below studies to your manuscript in discussion section using below manuscripts:

 DOI: 10.1155/2021/4946711

 DOI: 10.1155/2021/1520052

Ans. Thanks. I agree. I mention various 163methods of constructing modified exosomes as anticancer therapy. I mention advantages and disadvantages of these methods. Figure 4B also shows advantages and disadvantages of these methods.

I mention these studies (DOI: 10.1155/2021/4946711, DOI: 10.1155/2021/1520052): refs 163 and 157. I mention the role of mitochondria targeting: references 162, 163, lines 547-552. I mention the advantages of mitochondria targeting for development of anticancer therapy.

  • Q4. Conclusions should reaffirm the fundamental contribution of this paper.

Ans. In section 10 (conclusion and perspectives), I discuss advantages of using exosomes as delivery vehicles of anticancer molecules. I also mention future directions of studies concerning modified (artificial) exosomes. In this revision (conclusion and perspectives), I stress potential benefits of exosomes (artificial exosomes) as anticancer therapy.       

* In this revision, I sought professional help for English problems. I send English certificate. This effort might make this manuscript more readable.   

* In this revision, I change order of references to make this manuscript more readable.

* In this revision, I remove unnecessary and repetitive sentences to make this manuscript more readable. 

Round 2

Reviewer 1 Report

Dear authors,

thank you for making all the  recommended changes. In this form the review has now improved.

Author Response

Dear Sir. Thanks for your kindness. In this revision, I sought professional help for English problems. I send English certificate.

Sincerely yours

Jeoung Dooil

Reviewer 2 Report

The author's revision removes all my doubts and I think it is acceptable to accept and publish.

Author Response

Thanks for your kindness. In this revision, I sought professional help for English problems. I send English certificate.

Sincerely yours

Jeoung Dooil

Reviewer 3 Report

The authors revised according to the comments, but some places are still not acceptable:

  1. Figure 2-4: although the authors claimed, the font size of the text in the figure is still too small to read.
  2. Figure 2: why the size of the exosome is described as 40-160 nm, and how did others define it? Any comparison?
  3. Table 4: "Study design" should be "Prospective Outcome measures", and "Enrollment number" should be included as a separate column.
  4. The number of spaces before each paragraph is not consistent. Some paragraphs started with two spaces, some started with four spaces, and some started with five. Please revised the whole text to keep consistent.
  5. Line 301-303: "Since cancer cells secrete more exosomes than normal cells…" should include the reference.
  6. The aim of the study should be improved. You have described what you have done in the paper, but you have not presented the aim of the novelty/special aspects it brings to the field or the reason for choosing this topic. Please reshape the aim of the study.
  7. A paragraph or a separate section should be included to clearly describe how the literature was selected.

Author Response

Dear Sir

I thank for excellent suggestions. I made changes according to the suggestions. I hope that changes I made are fine. I send English certificate.

Sincerely yours

Jeoung Dooil

Q1. Figure2-4: although the authors claimed, the font size of the text in the figure is still too small to read.

Ans. Thanks. I change it. Please take look at new figures. 

Q2. Figure 2: why the size of the exosome is described as 40-160 nm, and how did others define it? Any comparison?

Ans. As far as size of the exosomes is concerned, it is highly controversial. Some reports show the size of exosomes is 30~150 nm while other reports show 40-~160nm [refs. 2-4]. Anyway, I change figure 2. Please take look at new figure 2. 

Q3. Table 4: "Study design" should be "Prospective Outcome measures", and "Enrollment number" should be included as a separate column.

Ans. Thanks. I add enrollment number. I change it as you suggested. Please take look at new table 4. 

Q4. The number of spaces before each paragraph is not consistent. Some paragraphs started with two spaces, some started with four spaces, and some started with five. Please revised the whole text to keep consistent.

Ans. Thanks. I change it as you suggested. Please take look at new manuscript.

Q5. Line 301-303: "Since cancer cells secrete more exosomes than normal cells…" should include the reference.

Ans. Thanks. I include reference (reference 2). Take look at new manuscript (lines 328-329). 

Q6. The aim of the study should be improved. You have described what you have done in the paper, but you have not presented the aim of the novelty/special aspects it brings to the field or the reason for choosing this topic. Please reshape the aim of the study.

Ans. I made changes to accommodate your suggestions. I add these sentences to accommodate your suggestions: Lines 52-53, 65-67, 133-134, 146-148, 159-163, 172-175, 193-195, 210-212, 225-227, 239-241, 251-252, 277-281, 289-290, 300-301, 318, 330-331, 342-343, 398-399, 435-437, 445-446, 453-456, 462-464, 481-483, 515-517, 544-546, 563-564, 606-607, 617-620, 626-627.

In this revision, I try to highlight aim of the study. I also delete unnecessary and repetitive sentences to make this manuscript more readable. I also change order of some sentences to make this manuscript more readable. Please take look at new manuscript.

Q7. A paragraph or a separate section should be included to clearly describe how the literature was selected.

Ans. Thanks. I agree. I add these sentences:

For this review, we first wanted to search papers concerning exosomes. More than 27,000 publication records were obtained through the PubMed search. A literature survey to identify papers describing properties and functions of exosomes was first conducted in PubMed on 5 December, 2022. For this review, we wanted to include both research papers and review papers. More than 90% of the papers in this review were published in the last 5 years. After removal of non-English publications, 163 publications were screened on title and abstract. For some papers (31/163), PDFs were not available. All these papers in this review are intended to enhance the understanding clinical values of exosomes, including biogenesis, functions, diagnostic and/or prognostic markers, and examples in drug delivery and therapy.

Please take look at new manuscript (lines 28-37).

* I search journals through PubMed and Scopus. I search articles and reviews that contain information relevant to exosomes. I choose a topic, decide on the scope of the review, and select database through PubMed and Scopus. I mostly search journals in the last 5-6 years.
